# Woman-Centered Care: Standardized Outcomes Measure

**DOI:** 10.3390/medicina59091537

**Published:** 2023-08-25

**Authors:** Milda Nagineviciute, Egle Bartuseviciene, Aurelija Blazeviciene

**Affiliations:** 1Department of Nursing, Faculty of Nursing, Lithuanian University of Health Sciences, 44307 Kaunas, Lithuania; aurelija.blazeviciene@lsmu.lt; 2Department of Obstetrics and Gynecology, Faculty of Medicine, Lithuanian University of Health Sciences, 44307 Kaunas, Lithuania; egle.bartuseviciene@lsmu.lt

**Keywords:** woman-centered care, midwife-led care, low-risk pregnancy, patient-reported outcomes, patient-reported experience measures

## Abstract

*Background and Objectives*: Patient- or woman-centered care, prioritizing women’s perspectives, needs, and preferences, is a widely recommended approach to enhance the quality of maternity care services. It aligns with the broader principles of patient-centered care, emphasizing the importance of a collaborative and respectful relationship between healthcare providers and women. This study evaluates low-risk pregnancies managed by midwives and obstetrician-gynecologists in Lithuania using patient-reported outcome measures and patient-reported experience measures. *Materials and Methods*: A prospective cohort study was conducted between September 2022 and April 2023. Data were collected through patient-reported questionnaires. *Results*: A total of 153 pregnant women who had singleton, low-risk pregnancies participated in the study, of whom 24.8% had their pregnancies supervised by a midwife, and 75.2% of the participants had their pregnancies supervised by an obstetrician-gynecologist. The study found no statistically significant differences in assessed patient-reported outcome measures and patient-reported experience measures between both models of care. *Conclusions*: Adopting patient-centered approaches enables healthcare systems to understand and address women’s specific needs and preferences, fostering high-quality and woman-centered care. This research adds to the growing evidence supporting midwife-led care and emphasizes the importance of personalized, woman-centered approaches in maternity care, ultimately enhancing maternal health outcomes and promoting positive experiences for low-risk pregnant women. The quality of care provided by midwives remains uncompromised and equivalently proficient compared to the care provided by collaborative teams.

## 1. Introduction

The concept of patient or woman-centered care, widely recommended in numerous countries globally, aims to enhance the quality of maternity care services by prioritizing women’s perspectives, needs, and preferences, thereby improving health outcomes, enhancing patient satisfaction, and promoting a positive healthcare experience for women [1,2,3,4]. It aligns with the broader principles of patient-centered care, emphasizing the importance of a collaborative and respectful relationship between healthcare providers and women [4,5].

Traditional maternity care measures used to evaluate the quality of care provided, such as mode of birth, post-partum hemorrhage, severe perineal trauma, induction of labor, and postnatal readmission, have limitations, as they primarily focus on clinical outcomes and processes [6,7,8]. Moreover, the current quality measures in pregnancy and childbirth care predominantly center on negative outcomes like morbidity and mortality, and although these are important outcomes even in high-income countries, they reveal little about the performance, quality, or value of maternity care from a woman’s perspective [6,7,8].

Women’s health and newborn health are critical indicators of a country’s overall economic health. The well-being and health status of women directly influences their participation in the workforce, productivity, and economic contributions. Investments in women’s health, including maternal care, and access to quality healthcare, have a substantial impact on reducing maternal and infant mortality rates, improving overall health outcomes, and enhancing the socio-economic development of a country [9].

Women’s health is receiving significant attention both internationally and nationally, as recognized in the Global Strategy for Women’s, Children’s, and Adolescents’ Health (2016–2030) by the World Health Organization (WHO) [10,11]. The Global Strategy for Women’s, Children’s and Adolescents’ Health emphasizes the need for comprehensive and integrated healthcare services for women, including maternal care, access to quality healthcare throughout their life course, and calls for a holistic approach that empowers women and ensures their access to essential healthcare services [10,11].

To address the limitations of traditional maternity care measures, there has been a growing recognition of the importance of woman-centered care and the integration of patient-reported outcome measures (PROMs) and patient-reported experience measures (PREMs) in evaluating the quality of maternity care [3,12,13].

By incorporating patient-reported outcome measures (PROMs) and patient-reported experience measures (PREMs) aligned with woman-centered care principles, healthcare systems could have a better understanding of and address the specific needs and outcomes that matter most to women [14].

Woman-centered care, which emphasizes the unique individual needs, expectations, and aspirations of women, is often associated with midwife-led care [15]. However, while studies in Lithuania have investigated the effects of midwife-led care and obstetrician-led care on maternal and neonatal outcomes in low-risk deliveries [16,17,18], there remains a research gap regarding the holistic evaluation of physical and emotional patient-reported outcomes experienced by women during low-risk pregnancies, from their own perspectives.

To address this gap, the incorporation of woman-centered care principles, and the use of patient-reported outcome measures (PROM) and patient-reported experience measures (PREM) in maternity care evaluation, offer a comprehensive approach to assessing the quality and value of care. By considering both clinical outcomes and patient-reported outcomes and experiences, healthcare systems can gain a more holistic understanding of the impact of care on women’s health and well-being. This patient-centered approach ensures that maternity care goes beyond focusing solely on negative outcomes and encompasses the woman’s perspective, ultimately promoting high-quality and patient-centered care [4,13]. Nevertheless, it is important to exercise caution when establishing such indicators, as variations are likely to exist among women, populations, and geographic areas, influenced by cultural and social norms [19].

The aim of this study was to assess and compare the PROMs and PREMs among women who received midwife-led care versus obstetrician-led care during low-risk pregnancies in Lithuania.

## 2. Materials and Methods

A prospective cohort study design was applied. The survey was conducted between September 2022 and April 2023. During the study period, a total of 164 questionnaires for women who had singleton, low-risk pregnancies and visited the designated healthcare facility during the research period were distributed (41 in the midwife-led (MW-led) group and 123 in the obstetrician-gynecologist-led (OB-led) group). Following the data collection phase, 153 questionnaires were considered eligible for analysis (n = 38 MW-led group, n = 115 OB-led group).

### 2.1. Research Context

In Lithuania, pregnant women have the option to receive care for low-risk pregnancies from a midwife, an obstetrician-gynecologist, or a family doctor [20]. According to established international midwifery standards and the prevailing Lithuanian medical guidelines, midwives possess the requisite competence to autonomously manage low-risk pregnancies, including the independent facilitation of low-risk deliveries. Despite these competencies, the scope of midwives’ authority within our domestic framework remains constrained. It is of paramount significance to substantiate the proposition that low-risk pregnancies can be capably managed with comparable effectiveness, achieving successful outcomes and ensuring safety, when entrusted to midwives operating autonomously. This proposition serves to contrast pregnancies overseen by midwives operating independently with those attended to by midwives collaborating within a team alongside an obstetrician-gynecologist.

To analyze the outcomes and experiences of low-risk pregnancies under different care models, the study focused on two distinct settings in Lithuania.

The university hospital represents a midwife-led approach to low-risk pregnancy care. In this setting, pregnant women receive comprehensive care from qualified midwives who independently manage and oversee the antenatal process. The regional hospital is another research setting that adopts an obstetrician-gynecologist-led approach to low-risk pregnancy care. Here, pregnant women receive their care from obstetrician-gynecologists, where midwives work as a team members. Both settings were selected as representative examples of different care models for the low-risk pregnancy care available in Lithuania.

Data collection would typically commence when a pregnant woman enters prenatal care, ideally during the first trimester (point 1), and continues between 28 and 34 weeks of gestation (point 2). However, if a woman initiates prenatal care closer to or within the third trimester, it may be appropriate to combine the data collection time points 1 and 2 as recommended by the International Consortium for Health Outcomes Measurement (ICHOM) [21]. The decision was made by the researchers to combine these two periods, as all terminated pregnancies before 22 weeks are classified as miscarriages.

Data for this study were collected through patient reports. Women with a low-risk pregnancy were asked to fill in a questionnaire after a standard antenatal care visit with the antenatal care provider. They were instructed to complete the questionnaires independently, ensuring privacy and confidentiality. The questionnaire was specifically completed during the period between 22 and 34 gestational weeks of pregnancy, ensuring data collection within a defined and consistent timeframe.

### 2.2. Research Instrument

For perinatal care, the ICHOM developed a patient-centered outcome set known as the Pregnancy and Childbirth (PCB) set. This set encompasses a combination of clinical measures and patient-reported measures, aiming to comprehensively evaluate the outcomes related to pregnancy and childbirth [22].

The questionnaire was designed according to the ICHOM Pregnancy and Childbirth Data Characteristics of Participants Collection Reference Guide [21].

With the authors’ concurrence, the questionnaire was linguistically adapted from English to Lithuanian, adhering to the ISPOR Principles of Good Practice: The Cross-Cultural Adaptation Process for Patient-Reported Outcome Measures, as prescribed in the ICHOM Pregnancy and Childbirth Data Characteristics of Participants Collection Reference Guide [21]. This entailed the initial translation from English to Lithuanian, the subsequent back translation from Lithuanian to English, and the iterative refinement of both English renditions until achieving complete congruence. The validation of the adapted instruments encompassed testing on a select cohort (pregnant women) of pertinent participants to evaluate alternative wording, gauge comprehensibility, elucidate interpretation nuances, and ascertain the cultural relevance of the translation.

In terms of substantive content, discernible disparities between the original instruments and the English-translated versions of the questionnaires did not emerge. As a consequence, the questionnaires were deemed suitable for implementation within the Lithuanian context.

Each questionnaire included seven PROM and two PREM measures (Table 1).

The PROMIS Global-10 questionnaire was used to assess patients’ health-related quality of life. It consists of 10 items with response options presented on 5-point rating scales. The questionnaire yields two summary scores: the Global Physical Health Score and the Global Mental Health Score, which are standardized using T-scores. Higher scores indicate better health [23].

For assessing anal incontinence, the Wexner Scale was utilized, comprising five items that evaluate the frequency and severity of fecal incontinence. To evaluate urinary incontinence, the ICQ-SF (International Consultation on Incontinence Questionnaire—Short Form) was used. This four-item questionnaire assesses the frequency, volume, and impact of urine leakage. Higher scores indicate greater severity [24].

To assess pain during intercourse, the PROMIS SFFAC102 questionnaire was used, tracking the impact of physical, emotional, and social factors on participants’ ability to engage in and enjoy intimate activities. Patient Health Questionnaire-2 (PHQ-2) was used to evaluate mental health. The main objective of the PHQ-2 is to conduct an initial screening for depression. It is a concise self-report questionnaire designed to identify depressive symptoms in individuals. Participants responded to two specific items using a 4-point Likert scale, where higher scores indicate a greater presence of depressive symptoms [25].

Breastfeeding confidence was assessed using the Breastfeeding Self-Efficacy Scale—Short Form (BSES-SF). Participants rated their agreement with a series of statements related to breastfeeding confidence on a Likert-type scale. Total scores were obtained by summing the item scores, with higher scores reflecting greater confidence in breastfeeding [26].

Each questionnaire included two PREM measures. The first measure assessed satisfaction with the results of care. The second measure evaluated shared decision-making, specifically addressing the confidence in care providers. This measure focused on the woman’s confidence as an active participant in decisions and her perceived confidence in healthcare professionals.

### 2.3. Statistical Analysis

The survey data were analyzed using the Statistical Package for Social Sciences (SPSS for Windows 29.0). Data are presented as absolute (n) and percentage frequencies (%) or means (m) ± standard deviation (SD), median (Md), and quartile range (Q1–3). Non-parametric methods of analysis were employed for variable comparisons. The Mann–Whitney U test was utilized to compare the distributions of quantitative variables between two independent samples. Correlation tables were constructed to assess the associations between attributes, while the chi-square (χ^2^) criterion was employed to determine attribute dependence. In cases where subgroup frequencies <5, Fisher’s exact criterion was utilized. To evaluate the strength of the association between PROMs and PREMs measures, the Spearman rank correlation coefficient was employed. A positive correlation coefficient indicated that as one variable increased, the other variable also increased, while a negative correlation coefficient indicated an inverse relationship. The strength of the relationship was assessed based on the magnitude of the correlation coefficient: a value between 0 and |≤0.3| was considered weak, between |>0.3| − |≤0.7| was considered moderate, and between |>0.7| − |1| was considered strong. The chosen level of statistical significance was set at *p* < 0.05.

### 2.4. Ethical Considerations

The study was approved by the Biomedical Research Ethics Committee of the Kaunas region (21 July 2022, No. P1-BE-2-24/2022). Participants had the freedom to decide on their participation in the study after informed consent was provided. Those who agreed to fill in the questionnaires signed the informed consent form. To ensure privacy and confidentiality, participants were given envelopes to place their completed questionnaire forms in. The sealed envelopes were collected by the research team at the healthcare facility. All collected data were handled in a secure manner, with strict adherence to ethical guidelines and data protection regulations.

## 3. Results

A total of 153 pregnant women participated in the study. Among the respondents, 24.8% had their pregnancies supervised by a midwife (MW-led), while the majority, comprising 75.2% of the participants, had their pregnancies supervised by an obstetrician-gynecologist (OB-led). The study participant characteristics were found to be similar in both groups (Table 2).

A statistical analysis was performed to analyze the differences between the MW-led and OB-led groups in terms of PROMs and PREMs measures. The results of the study revealed that there were no statistically significant differences in the assessed PROMs and PREMs between both models of care (Table 3).

The associations between the standardized PROMs and PREMs were analyzed. The results of the correlation analysis revealed statistically significant weak and moderate correlation coefficients between these measurements (Table 4).

Better physical health was associated with better mental health. Better mental health was linked to increased self-confidence in caring for the baby, higher confidence in breastfeeding, and a weaker impact of pain on sexual life. A higher self-confidence in baby care was associated with increased confidence in breastfeeding. More severe anal incontinence demonstrated a positive correlation with more frequent psychological problems and an increased severity of urinary incontinence.

## 4. Discussion

Standardized indicators play a crucial role in improving healthcare quality, enhancing patient satisfaction with care, and assessing various aspects of healthcare services. The vital importance of standardized indicators in healthcare encompasses various fields, including maternal and child health. The implementation of standardized indicators has a huge value as it ensures that patients receive safe and high-quality services, irrespective of their country of residence or cultural context. Setting equal indicators for evaluating healthcare outcomes and experiences from patients’ perspectives, standardized indicators offer a powerful tool for consistent and evidence-based practices [27,28,29,30].

Assessing the quality of midwifery care, a considerable number of studies have primarily centered on birth and the postnatal period. These studies have demonstrated that patient satisfaction with care and outcomes does not significantly differ between midwife-led care and the other care group [31]. Moreover, in certain instances, care provided by midwives has been associated with a higher satisfaction with care and improved outcomes [32,33,34,35]. Despite the wealth of research on midwifery care during birth and the postnatal period, the focus on antenatal care has been comparatively scarce [36]. Our research findings highlight the similar effectiveness and quality of care provided by both midwives and obstetrician-gynecologists in the context of the assessed measures. It suggests that both care models can deliver satisfactory results and positive experiences for pregnant women. As antenatal care is a critical phase in the maternity journey, further research in this area is warranted to gain a comprehensive understanding of the quality and outcomes of midwifery-led antenatal care.

Midwife-led care holds paramount importance as it embodies a patient-centered approach, taking into consideration the care from the woman’s perspective [37]. During pregnancy, the focus on health-related quality of life is crucial, encompassing not only physiological changes but also overall well-being [38]. Our study revealed that both physical and mental health achieved high scores in both the midwife-led care and obstetrician-gynecologist-led care groups, indicating positive outcomes for maternal health. The outcomes of our study align with those observed in a previous study [39]. However, this research identified that the scores exhibited a broader range, encompassing both the upper and lower boundaries of the physical and mental health domains. While our study revealed high scores in both physical and mental health, the broader range observed in the other study suggests that individual experiences may vary widely within different care settings.

Research has shown that maternal mental health during pregnancy can have long-lasting effects on both the mother and the child [40]. The influence of mental health extends beyond the antenatal period and can impact decisions made after birth, such as the choice to breastfeed. Our findings revealed a positive association between better mental health and an increased confidence in breastfeeding, highlighting the role of maternal well-being in promoting successful breastfeeding practices. The research indicates that providing appropriate education and support during the prenatal period can positively impact a woman’s confidence and ability to breastfeed after birth [41,42]. Moreover, our study identified that higher self-confidence in baby care was correlated with an increased confidence in breastfeeding. This suggests that maternal self-assurance in caring for the newborn plays a role in fostering positive breastfeeding experiences. Building confidence during pregnancy care is also vital as it can significantly influence a woman’s self-assurance as a mother. By prioritizing comprehensive and supportive care during pregnancy, healthcare providers can positively impact women’s maternal confidence, thus contributing to improved maternal well-being and postnatal experiences.

The concept of woman-centered care is significantly highlighted within the context of our study. One of the 4 PROM domains, namely the Patient Satisfaction with Care assessment, is inherently intertwined with the core tenets of woman-centered care. This domain extends beyond mere contentment with the provided service; it encapsulates the engagement in collaborative decision-making with healthcare professionals on matters pertaining to health. Extensive research underscores that patient satisfaction with care serves not solely as a process metric reflecting care quality, but also manifests a positive correlation with patient-reported outcomes [43]. As such, patient satisfaction wields a direct influence in shaping the trajectory of patient-centered care, a pivotal aspect of woman-centered care [43,44,45].

In the realm of practice, these indicators become essential, particularly in addressing the challenges posed by the hesitance of healthcare practitioners towards the adoption of patient engagement in shared decision-making within clinical settings. To this end, the comprehensive conceptualization and meticulous measurement of shared decision-making could potentially furnish a more robust foundation of evidence, subsequently guiding the implementation of woman-centered care principles.

In the pursuit of a woman-centered care model, a crucial step involves investing in midwife-led care. This imperative message serves as a directive to policymakers, underscoring the necessity of providing maternity care on a comprehensive basis, empowering midwives to assume more autonomous roles. By promoting midwife-led care, healthcare systems can adopt a woman-centered approach that prioritizes the individual needs and preferences of pregnant women, promoting an environment of personalized and woman-centered care [46,47].

### Strengths and Limitations

The strength of our study lies in its pioneering approach as the first in the country to evaluate standardized indicators in maternity care for the measurement of PROM and PREM. This innovative use of standardized indicators allows for a comprehensive and uniform assessment of the key aspects of woman-centered care. Furthermore, the study’s results will enable us to assess the quality of maternity care from a national perspective, providing valuable insights into the strengths and areas for improvement within the country’s midwife-led care system. Additionally, by comparing our findings with other countries where midwife-led care is already the dominant care model, our study offers a unique opportunity to gain international perspectives on maternal healthcare practices and outcomes. By adopting this methodology and combining national and international comparisons, our research contributes to the advancement of evidence-based midwife-led care practices, with the potential to shape policies and interventions that enhance maternal and neonatal well-being locally.

While our study has yielded valuable insights, it is essential to acknowledge its inherent limitations. One notable limitation is the constrained availability of midwives practicing as independent, full-scope specialists in all healthcare institutions across the country. Consequently, our study had to contend with a relatively small sample size, potentially limiting the generalizability of our findings to the entire population. Furthermore, the country’s current state of understanding midwife-led maternity care poses a challenge to the comprehensive assessment of woman-centered care. As the country is still embracing and defining the role of midwife-led care in maternal healthcare, our study’s scope may not encompass the full spectrum of woman-centered care practices.

## 5. Conclusions

The study highlights the importance of woman-centered care principles and the integration of patient-reported measures in evaluating maternity care quality. By adopting patient-centered approaches, healthcare systems can better understand and address the specific needs and preferences of women, ultimately promoting high-quality and patient-centered care. This study contributes to the growing body of evidence supporting midwife-led care and highlights the importance of personalized, woman-centered approaches in maternity care settings. By prioritizing women’s perspectives and preferences, healthcare systems can enhance maternal health outcomes and promote positive experiences for pregnant women during low-risk pregnancies. Hence, our study underscores the imperative retention of midwives as primary caregivers for low-risk pregnancies, in alignment with established medical standards. This affirmation is rooted in the finding that the quality of care provided by midwives remains uncompromised and equivalently proficient compared to the care provided by collaborative teams.

## Figures and Tables

**Table 1 medicina-59-01537-t001:** PROM and PREM structures.

		Validated Scoring Tool	No of Questions	Score (Min–Max)	Threshold for a Suboptimal Score	Interpretation Score
PROMs	Health-related quality of life	PROMIS-10	10	PH 29.6–67.7; MH 21.2–67.6	PH < 42; MH < 40	↑ Quality of life
	Urinary incontinence	ICIQ-SF	4	0–21	≥6	↑ Severity
	Anal incontinence	Wexner	5	0–20		
	Pain with intercourse	PROMIS SFFAC102	1	1–5		↑ Impact of pain
	Mental health	PHQ-2	2	0–6	≥3	↑ Depression risk
	Confidence in breastfeeding	BSES-SF	14	14–70		↑ Confidence
	Role confidence	N/A	1	1–5		↑ Confidence
PREMs	Satisfaction with care	N/A	1	1–5		↑ Satisfied
	Healthcare responsiveness and shared decision making	N/A	4	0–2		↑ Better experience

BSES-SF, Breastfeeding Self-Efficacy Scale—Short Form; N/A, not available; PROM, patient-reported outcome measure; PREM, patient-reported experience measure; PROMIS, Patient-Reported Outcome Measurement Information System; PH, physical health; MH, mental health; PHQ-2, patient health questionnaire; ICIQ-SF, International Consultation on Incontinence Questionnaire—Short Form; SFFAC102, sexual function and satisfaction. ↑—the Increasement of the Interpretation Score.

**Table 2 medicina-59-01537-t002:** Characteristics of the participants.

Characteristics	MW-Led (n = 38)	OB-Led (n = 115)	*p*
Maternal age (mean ± SD)	30.5 (4.1)	29.4 (4.0)	0.126
Educational level			
Primary/Secondary/Vocational	5 (13.2)	22 (19.1)	0.446
Higher Non-University	4 (10.5)	18 (15.7)	
Higher University	29 (76.3)	75 (65.2)	
Marital status			
Married	30 (78.9)	89 (77.4)	0.901
Cohabitation	7 (18.4)	24 (20.9)	
Single/Divorced	1 (2.6)	2 (1.7)	
Parity			
Nulliparous	28 (73.7)	72 (62.6)	0.214
Multiparous	10 (26.3)	43 (37.4)	
Pre-pregnancy body mass index (kg/m^2^)			
<18.5	0	6 (5.2)	0.419
18.5–24.9	26 (68.4)	80 (69.6)	
25.0–29.9	8 (21.1)	22 (19.1)	
≥30	4 (10.5)	7 (6.1)	
Smoking before pregnancy	3 (7.9)	23 (20.0)	0.219

Values are number (percentage) unless otherwise stated.

**Table 3 medicina-59-01537-t003:** Comparison of PROMs and PREMs between MW-led and OB-led groups.

Characteristics		MW-Led (n = 38)		OB-Led (n = 115)		
		m (SD)/ n (%)	Median (Range)	m (SD)/ n (%)	Median (Range)	*p*
Health-related quality of life (T score)	PH	45.5 (5.0)	46.3 (42.3–47.7)	46.6 (4.6)	47.7 (44.9–50.8)	0.344
	PH *	4 (10.5)		14 (12.2)		0.785
	MH	51.31 (4.7)	52.1 (48.3–53.3)	51.75 (7.2)	50.8 (45.8–59.0)	0.860
	MH **	1 (2.6)		5 (4.3)		0.637
Incontinence	ICIQ-SF	0.84 (2.0)	0 (0–0)	1.15 (2.7)	0 (0–0)	0.860
	ICIQ-SF ***	7 (18.4)		22 (19.1)		0.923
	Wexner	0.63 (1.9)	0 (0–1)	0.79 (1.4)	0 (0–1)	0.218
	Wexner ***	10 (26.3)		41 (35.7)		0.290
Pain with intercourse		2.47 (1.3)	2 (1–4)	2.17 (1.1)	2 (1–3)	0.243
Mental health screening	PHQ-2	1.3 (1.3)	1.0 (0–2)	1.0 (1.1)	1.0 (0–2)	0.125
	No depressive disorder	35 (92.1)		107 (93.0)		0.846
	Major depressive disorder	3 (7.9)		8 (7.0)		
Confidence in breastfeeding		49.5 (9.8)	48.5 (42.7–55.3)	48.9 (10.3)	50.0 (43.0–56.0)	0.919
Role confidence		3.9 (0.8)	4 (4–4)	1.0 (0.8)	4 (4–4)	0.777
Satisfaction with care		4.2 (1.1)	4.5 (4–5)	4.4 (1.2)	5 (4–5)	0.146
Healthcare responsiveness and shared decision-making		1.8 (0.4)	2.0 (1.8–2.0)	1.9 (0.3)	2.0 (1.8–2.0)	0.559

* T-standardized scores < 42; ** T-standardized scores < 40; *** answered positively to at least one question.

**Table 4 medicina-59-01537-t004:** Associations between PROMs and PREMs indicators (Spearman correlation, r).

Measurements	1	2	3	4	5	6	7	8	9
Physical health (1)	-								
Mental health (2)	0.54 **	-							
Depressive disorder (3)	−0.35 **	−0.48 **	-						
Pain with intercourse (4)	−0.17 *	−0.20 *	0.25 **	-					
Role confidence (5)	0.16	0.23 *	−0.20 *	−0.09	-				
Severity of urinary incontinence (6)	0.01	−0.06	0.04	−0.06	0.07	-			
Severity of anal incontinence (7)	−0.15	−0.03	0.15	0.10	0.07	0.18 *	-		
Confidence in breastfeeding (8)	0.18 *	0.19 *	−0.17 *	−0.04	0.40 **	0.02	−0.01	-	
Satisfaction with care (9)	0.16 *	0.10	−0.02	0.07	0.09	0.06	0.09	−0.02	
Healthcare responsiveness and shared decision-making (10)	0.07	0.06	−0.13	−0.07	−0.07	0.02	−0.07	0.10	0.09

* *p* < 0.05, ** *p* < 0.001. 1–10—PROMs and PREMs indicators.

## Data Availability

The data presented in this study are available on request.

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
