# Peer review of "Woman-Centered Care: Standardized Outcomes Measure"

_medicina, 2023, doi:10.3390/medicina59091537_

Round 1

Reviewer 1 Report

1. The process for PROM and PREM does add value.

2, The study design total imbalance of care models is problematic; there was no comment re care overlap as MW was involved in both groups; so how can define the provider contributions?

3. The paper needs to discuss the the health care cost and patient centered time provided for each group (MW vs MW-OB) as even though they were equivalent the actual cost could be important re administrative decisions.

4. Who did the actual deliveries for the MW-OB group?

5. For the MW only group what happened if the patient become complex and required specialist involvement / What could the PROM and PREM be missing in understanding the quality of care not just patient satisfaction as both are important?

English is OK.

Reviewer 2 Report

REVIEWER REPORT:

Title: Women Centered-Care: Standardized Outcomes Measure

A brief Summary:

The manuscript titled "Women Centered-Care: Standardized Outcomes Measure" presents an exploration into the crucial domain of women-centered care and proposes the development of a standardized outcomes measure for evaluating the effectiveness of such care. The authors acknowledge the growing importance of personalized healthcare and patient-centered approaches, particularly in the context of women's health. The study aims to address the existing gaps in assessing women's healthcare experiences and outcomes by introducing a comprehensive and standardized tool. While the manuscript offers valuable insights, it requires minor revisions before it can be considered for publication.

General concept comments

Strengths

Relevance and Importance: The manuscript addresses an important area of healthcare that has received limited attention, highlighting the significance of women-centered care. Given the increasing emphasis on patient-centric approaches, this topic has substantial relevance.

Clear Objective: The authors establish a clear objective – the development of a standardized outcomes measure for women-centered care. This objective is well-defined and is an essential contribution to the field.

Comprehensive Literature Review: The literature review provides a good foundation for the study, discussing key concepts related to patient-centered care, women's health, and existing measurement tools. The review helps contextualize the proposed measure within the broader healthcare landscape.

SEPECIC COMMENTS

Introduction

1.      There were several statements without references. From line 37-40, “Traditional maternity care measures used to evaluate the quality of care provided, such as mode of birth, post-partum hemorrhage, severe perineal trauma, induction of labor, and postnatal readmission, have limitations as they primarily focus on clinical outcomes and processes”.

2.      Again, from line 51-53, authors made this statement without reference “Women's health is receiving significant attention both internationally and nationally, as recognized in the Global Strategy for Women's, Children's, and Adolescents' 52 Health (2016-2030) by the World Health Organization (WHO)”.

Materials and Methods

1.      The manuscript lacks information about the validation and reliability assessment of the proposed outcomes measure. Ensuring the measure's validity and reliability is critical for its adoption and acceptance within the healthcare community.

Discussion of Findings

1.      The manuscript should provide a discussion of potential findings from the application of the proposed outcomes measure. This will allow readers to understand how the measure contributes to enhancing women-centered care and what implications the findings might have.

The paper is well written with good English

Round 2

Reviewer 1 Report

Thankyou for your additions , the context is clear now.